# Feature Interaction-Based Reinforcement Learning for Tabular Anomaly Detection

Yaoxun Liu, Liangli Ma *, Muyuan Wang and Siyuan Zhang

College of Electronic Engineering, Naval University of Engineering, Wuhan 430033, China; lyx_nue@163.com (Y.L.)
* Correspondence: maliangl@163.com

**Abstract:** Deep learning-based anomaly detection (DAD) has been a hot topic of research in various domains. Despite being the most common data type, DAD for tabular data remains under-explored. Due to the scarcity of anomalies in real-world scenarios, deep semi-supervised learning methods have come to dominate, which build deep learning models and leverage a limited number of labeled anomalies and large-scale unlabeled data to improve their detection capabilities. However, existing works share two drawbacks. (1) Most of them simply treat the unlabeled samples as normal ones, ignoring the problem of label contamination, which is very common in real-world datasets. (2) Only very few works have designed models specifically for tabular data instead of migrating models from other domains to tabular data. Both of them will limit the model's performance. In this work, we propose a feature interaction-based reinforcement learning for tabular anomaly detection, FIRTAD. FIRTAD incorporates a feature interaction module into a deep reinforcement learning framework; the former can model tabular data by learning a relationship among features, while the latter can effectively exploit available information and fully explore suspicious anomalies from the unlabeled samples. Extensive experiments on three datasets not only demonstrate its superiority over the state-of-art methods but also confirm its robustness to anomaly rarity, label contamination and unknown anomalies.

**Keywords:** deep reinforcement learning; anomaly detection; semi-supervised learning; feature interaction; tabular data





## 1. Introduction

Tabular data refers to data that are arranged in the form of a table, in which each row represents a sample, and each column represents a feature. As the most common type of data in real-world applications, tabular data are widely used in many domains, such as network security [1,2], financial transaction [3,4], industrial manufacturing [5,6], marine traffic-cite [7,8], etc. Anomalies (also called outlier or novelty), which exist in almost all domain applications, often indicate malfunctions or malicious behavior and may result in property damage or even casualties. Anomaly detection (AD) for tabular data has been a lasting yet active topic in the last few decades, and dozens of methods have been proposed for different tasks.

Due to the rarity of anomalies, most real-world datasets are severely imbalanced, i.e., negative instances (normal samples) account for the vast majority, while positive instances (anomalous samples) account for only a small minority. Therefore, many researchers consider anomaly detection as an unsupervised learning problem, such as proximity-based methods [9–11], ensemble-based methods [12–14], and neural network-based methods [15–17]. In the last decade, a few studies [18–20] have pointed out the availability of labeled anomalies in some real-world scenarios. A limited number of positive samples that come from the identification of experts or the accumulation of the system are usually available with trivial cost. In recent years, more and more works have considered anomaly detection as a semi-supervised or weakly-supervised learning problem,

leveraging a small set of labeled anomalies and a large-scale unlabeled dataset to train AD models. Unfortunately, most existing works [18–22] treat all the samples from the unlabeled dataset as normal for convenience and overlook possible anomalies (also called anomalous contamination). Considering the very large data size, even if the proportion of anomaly samples is extremely low, there would exist quite a few anomalies in the unlabeled dataset. Ignoring these samples may result in a loss of information and then limitations in a model's performance.

Additionally, most existing deep models for tabular data anomaly detection (TAD) [18–20,22,23] try to transfer AD approaches from other domains, such as computer vision (CV) or natural language processing (NLP), instead of modeling the characteristics of tabular data. Studies [24,25] indicate that deep models that excel in CV or NLP cannot achieve the desired performance on tabular data due to its characteristics, including lack of locality, data sparsity and mixed feature of types. The most notable difference between tabular data and other types of data is the associative relationship between columns. Practices [26–29] in Click-Through prediction (CTR) demonstrate that feature interactions, especially high-order feature interactions, are crucial to modeling tabular data. However, to the best of our knowledge, no paper has yet worked on how to apply feature interaction to TAD.

To cope with the problem of anomalous contamination in an unlabeled dataset, we propose a novel method based on deep reinforcement learning (DRL). As can be seen from the name, DRL combines the expression ability of deep learning and the decision-making ability of reinforcement learning (RL). Different from unsupervised learning and semi-supervised learning methods, RL updates parameters by interacting with the environment without requiring data to be given in advance. In this work, we leverage the DRL algorithm to train an anomaly detector that can not only fit the labeled anomalies but also detect possible anomalies from the unlabeled dataset.

To settle the problem that deep models do not model tabular data well, we introduce a tabular data modeling approach named gated adaptive feature interaction network (GAIN) [29]. GAIN exploits multiple parallel interaction units to learn useful high-order feature interactions. The parallel design guarantees a high-efficiency model, as works [30,31] have experimentally demonstrated that parallel architectures can dramatically reduce the processing time of models both on CPUs and on GPUs. GAIN works as a middleware, which can transform raw features into informative representations and can replace the deep module in DRL.

We further instantiate the proposed approach into a model called feature interaction-based reinforcement learning for tabular data anomaly detection (FIRTAD), and the architecture is shown in Figure 1. We choose the soft Actor–Critic (SAC) [32] as the main framework of the FIRTAD. The policy network and the Q-network share the same deep module, which is implemented with GAIN. We create a simulation environment to interact with the SAC, which includes labeled anomalies and an unlabeled dataset. To ensure the exploitation of all samples, we propose a novel sampling strategy that prevents repeated sampling from a densely distributed region. To encourage the agent to explore samples, which can bring more novelty, we extend the reward function with an intrinsic reward. More details are discussed in Section 3.

We summarize the main contributions of this work as follows:

1. We propose a novel DRL-based anomaly detection approach specifically for tabular data and deliberately devise a simulation environment that allows all samples to be fully explored.
2. We introduce a feature interaction module (GAIN) into our approach, which can model the characteristics of tabular data by learning interactions between features. To the best of our knowledge, it is the first effort to apply feature interactions to anomaly detection.
3. We instantiate the proposed approach into a model called FIRTAD and extensively evaluate the model, comparing six baselines on three benchmark datasets. The

experimental results demonstrate that our model performs better than state-of-the-art models and exhibits better robustness to class imbalance, label contamination and unknown anomalies.

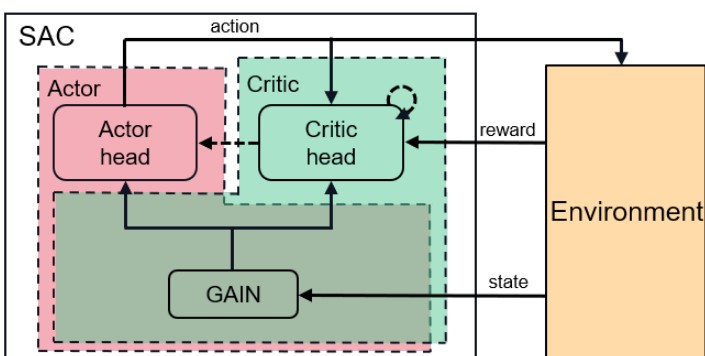

**Figure 1.** Architecture of FIRTAD. FIRTAD, which is based on a deep reinforcement learning framework, consists of two parts, the agent and the environment. The agent is implemented by an SAC algorithm, in which the Actor and the Critic share the same feature interaction module GAIN as their deep modules. The environment is implemented with a sampling function and a reward function.

## 2. Related Works

### 2.1. Anomaly Detection Methods

From the perspective of the availability of supervision information (labels), most existing AD methods can be divided into three categories: unsupervised learning, supervised learning and semi-supervised (weakly supervised) learning methods. Due to the high cost of data-labeling processes in real-world scenarios, supervised learning methods are often impractical, and the other two are much more popular.

#### 2.1.1. Unsupervised Anomaly Detection

Unsupervised AD methods are almost based on the assumption that normal samples have different distributions from anomalous samples. Normal samples are densely distributed, while anomalous samples are sparsely distributed and far from normal ones. Some works [9,10,33] treat samples as data points and detect anomalies by calculating the distance or density; Other works [11–13] judge the degree of abnormality through comprehensive analysis of the distribution of every single dimension. Ref. [34] obtains the anomaly scores of samples by calculating the joint distribution of all dimensions. Conventional machine learning methods fail to work effectively when dealing with high-dimensional samples due to the curse of dimensionality. To tackle this problem, deep learning methods are introduced to anomaly detection; the common practice is to exploit deep neural networks (e.g., multi-layer perception (MLP), autoencoder (AE) or generative adversarial network (GAN)) to project samples into a low-dimensional representation space, and then distinguish anomalies from normal samples [15,35,36]. Due to the lack of supervised information, almost all unsupervised methods detect anomalies by modeling normality. Although the unsupervised AD methods have achieved decent results, their performances are still limited because anomalies that can be easily obtained have not been utilized.

#### 2.1.2. Supervised Anomaly Detection

Although supervised AD methods are not as popular as unsupervised and semi-supervised ones, they still attract the attention of many researchers. Traditional supervised methods treat AD as a binary classification problem, and commonly used methods include Naive Bayesian, Support Vector Machine (SVM) and Gradient Boosting Decision Tree (GBDT). Deep learning-based methods train a classifier to detect anomalies on the basis of representation learning; well-known works include MLP, ResNet and FTTransformer [24]. Both traditional and deep methods have their own drawbacks. Traditional methods do not

perform well when processing high-dimensional, heterogeneous or non-independent data. In contrast, the deep methods, although they can handle these problems well, require a large number of labeled samples for training.

### 2.1.3. Semi-Supervised Anomaly Detection

Semi-supervised AD methods utilize limited supervised information to improve the ability to identify anomalies. Some works leverage labeled anomalies to enhance existing unsupervised AD models. Ref. [18] uses supervised information to push anomalies away from the center of a compact hypersphere, which is built using an unsupervised method [35]. Ref. [22] introduces anomalous samples to a distance-based method (e.g., K-nearest neighbors [10]) and identifies anomalies by calculating the knn-distance between the query sample and a random unlabeled subset. Other works introduce labeled anomalies into supervised AD models, and solve the problem of class imbalance through data augmentation, downsampling, etc. Ref. [37] proposes two strategies to enrich the anomalous samples and distinguishes anomalies using a contrastive learning method. Ref. [19] builds instance pairs to make the proportion of instance pairs containing anomalies reach 50%. Refs. [20,21] train their models on datasets that are equally sampled from both labeled anomalies and unlabeled samples. Ref. [38] leverages anomalies to obtain a prior anomaly score for each sample and uses the score as supervised information to optimize the AD model. Due to the exploitation of labeled anomalies, semi-supervised methods significantly improve the performance of AD models. However, almost all semi-supervised AD models are based on the assumption that the unlabeled subset contains normal samples only or do not consider the impact of possible anomalies in it.

### 2.2. Feature Interaction of Tabular Data

Due to the lack of locality and the complexity of features, tabular data cannot be modeled well with prevailing deep models, such as MLP, convolutional neural network (CNN), recurrent neural network (RNN), etc. Most deep anomaly detection approaches project high-dimensional tabular data into a low-dimensional space using MLP, ResNet, AE or GAN. However, these architectures cannot guarantee the preservation of discriminative information because they overlook feature interaction, which is the most prominent difference between tabular data and other data types.

Feature interaction, especially high-order feature interaction, has been proven to be crucial in improving the model's expressiveness in CTR prediction tasks [26–28,39]. In [40], Rendle argues that a second-order feature interaction can be represented with the inner product of two latent vectors, each of which represents a single feature, and propose factorization machine (FM) to automatically learn all possible second-order feature interactions. Many works [26–28] extend FM to learn higher-order feature interactions. However, these works brutely enumerate all possible feature interactions without differentiating their importance, and the introduction of useless interactions not only increases the computational complexity but also downgrades the model's performance. Xue et al. [41] propose AutoHash to adaptively learn useful high-order interactions. In AutoHash, all features are put into $k$ buckets with randomly initialized probabilities (features can be reused), and every bucket represents a feature interaction. The probabilities are learnable variables that help the buckets preserve useful interactions through training. Liu et al. [42] propose a two-stage algorithm called automatic feature interaction selection (AutoFIS). In the first stage, the model is trained to drop interactions that contribute little to the final prediction; in the second stage, the model is re-trained to learn the importance pf the retained interactions. Chen et al. [43] propose a bayesian higher-order feature interaction selection (BH-FIS). BH-FIS implements the enumeration of all feature interactions by using outer-product and masking techniques and employs spike-and-slab priors to distinguish useful feature interactions from useless ones. Liu et al. [29] propose a gated adaptive feature interaction network (GAIN) that can adaptively learn high-order feature interactions. GAIN consists of a cross-module and a deep module; the former exploits multiple parallel interaction

units to explicitly model feature interactions, while the latter leverages an MLP to model feature interactions in an implicit way.

Although the effectiveness of feature interactions has been proven in modeling tabular data, surprisingly, we can hardly find a work that incorporates it into tabular data anomaly detection.

*2.3. Deep Reinforcement Learning for Tabular Anomaly Detection*

The vanilla DRL algorithm is suitable for time series data due to its dependence on a live environment. Lopez-Martin et al. [44] make a conceptual modification of the vanilla DRL algorithm to make it feasible for tabular data, replacing the environment with a sampling function and designing a reward function based on the detection error. In addition, the authors make a comparison of several DRL algorithms on network intrusion detection datasets. Vimal et al. [3] exploit a DQN to tackle the payment fraud detection problem and use the technology of experience replay to improve the efficiency of sampling.

To solve the AD problem with a small set of labeled anomalies and a large-scale unlabeled dataset, Pang et al. [45] propose an approach called Deep Q-learning with Partially Labeled ANomalies (DPLAN). DPLAN creates an anomaly-biased simulation environment that continuously samples anomalies or suspected anomalies from the whole dataset. Separate sampling functions are designed for the labeled anomaly set and the unlabelled dataset, denoted as $g_a$ and $g_u$, respectively. $g_a$ selects samples uniformly from the labeled anomaly set, while $g_u$ selects samples that are likely to be anomalous from the unlabeled dataset. $g_u$ is defined as

$$g_u(s_{t+1}|s_t, a_t) = \begin{cases} \underset{s \in S^u}{\arg\min}\, d(s_t, s) & if\ a_t = 1 \\ \underset{s \in S^u}{\arg\max}\, d(s_t, s) & if\ a_t = 0, \end{cases} \tag{1}$$

where $S^u$ denotes a random subset of the unlabeled dataset, $d$ denotes a function of Euclidean distance and $a_t$ denotes the agent's judgment on $s_t$. $a_t = 1$ means the agent identifies $s_t$ as an anomaly, the sample nearest to $s_t$ is considered most likely to be anomalous, and is selected as $s_{t+1}$. In contrast, $a_t = 0$ means $s_t$ is identified as a normal sample, and the sample farthest from $s_t$ is returned to the agent.

DPLAN also designs a combined reward function $r_t = r_t^e + r_t^i$. The external reward $r_t^e$ is defined based on the prediction error. $r_t^e$ will return a positive reward if an anomalous sample is correctly recognized by the agent, no reward will be returned if a normal sample is correctly identified, and a negative reward will be returned if the agent makes a mistake. The external reward function is defined as

$$r_t^e = \begin{cases} 1 & if\ a_t = 1\ and\ s_t \in D^a \\ 0 & if\ a_t = 0\ and\ s_t \in D^u \\ -1 & otherwise, \end{cases} \tag{2}$$

where $D^a$ and $D^u$ denote the labeled set and the unlabeled dataset, respectively.

The intrinsic reward $r_t^i$ is devised to encourage the agent to explore novel anomalies. Hence, samples from lower-density regions receive higher intrinsic rewards as DPLAN assumes that anomalous samples are sparsely distributed and far from normal ones. The iForest [12] algorithm can indicate the degree of abnormality of a sample and is, therefore, used to calculate the intrinsic reward. The intrinsic reward function is defined as $r_t^i = iForest(s_t)$.

Due to the special design of the anomaly-biased sampling function and the combined reward, the agent of DPLAN is encouraged to explore the unlabeled dataset for possible anomalies. However, the DPLAN still has three main drawbacks. First, the discrete action space, i.e., $A = \{0, 1\}$, severely limits the model's performance. The identification ability of the agent is gradually improved through training. Before the model converges, the agent's actions are usually undetermined. However, the discrete action space cannot express the

uncertainty accurately, and the step changes between 0 and 1 would hinder the model from learning useful information. Secondly, the external rewards are designed to be too coarse-grained to cover all situations, e.g., when the agent takes an ambiguous action. Thirdly, the anomaly-biased sampling function may result in excessive exploration of anomalies and insufficient exploration of normal samples, which will be sub-optimal.

## 3. Our Proposed Method

### 3.1. Problem Definition

Given a training dataset $D = D^a \cup D^u$, where $D^a$ and $D^u$ denote a small labeled anomalous subset and a large-scale unlabeled subset, respectively. The size of $D^a$ is much smaller than that of $D^u$, and the ratio of their sizes usually does not exceed 10%. The vast majority of the samples in $D^u$ are normal, and only a very small number of samples are anomalous, part of which may come from unknown classes (i.e., classes that have not appeared in $D^a$). We aim to find out hidden anomalies from $D^u$ by taking full advantage of the whole dataset $D$. Note that $D$ is a tabular dataset in which all samples are independent of each other, and there is no temporal relationship between samples.

To apply a DRL algorithm to anomaly detection, we formulate the binary classification problem as a sequential decision-making problem. The agent receives a sample $s_t$ from the environment at time $t$, and takes an action $a_t$. The environment gives a reward $r_t$ and a new sample $s_{t+1}$ to the agent according to $s_t$ and $a_t$. The interaction between the agent and the environment can be represented with a Markov Decision Process (MDP), which is defined as below:

- **State space**. The whole dataset $D$ (including unlabeled dataset $D^u$ and labeled anomalies $D^a$) is defined as the state space. Each $s_t \in D$ denotes the state received from the environment at time $t$.
- **Action space**. Different from the existing works, we define a continuous action space $A = [0, 1]$. Therefore, the action $a_t$ can also be regarded as the anomaly probability of $s_t$. The closer $a_t$ is to 1, the more likely $s_t$ is an anomaly and vice versa.
- **State transition**. After the agent takes an action $a_t$, the environment renders a new state $s_{t+1}$ to the agent. Different from the anomaly-biased strategy used by DPLAN, which is dedicated to sampling anomalous states, we propose a novel sampling strategy that fully explores the entire data space.
- **Reward**. Similar to DPLAN, our proposed method leverages a combined reward function. We design a continuous extrinsic reward function to be compatible with the continuous action space. When $s_t$ comes from $D^a$, the agent will obtain a large penalty if it fails to recognize $s_t$. When $s_t$ comes from $D^u$, identifying $s_t$ as an anomaly should not be given a large penalty as anomalies may be hidden in $D^u$. In addition, we design a curiosity-driven intrinsic reward function to encourage the agent to explore the entire state space. The reward function is defined as $r_t = r_t^e + \lambda r_t^i$, where $\lambda$ is a scalar weighting the relevance of the intrinsic reward, and it takes a value from $[0, 1]$.

### 3.2. Agent

#### 3.2.1. Foundation of the Proposed Approach

The agent of our proposed model is implemented with a SAC, which is a stochastic policy algorithm that can deal with continuous action space. Different from other DRL algorithms that aim to learn a policy to maximize the cumulative rewards, the SAC augments the objective with an entropy regularization term to concurrently maximize the entropy of the agent's action. The introduction of the maximum entropy can not only promote the exploration but also prevent premature convergence. The objective of the SAC is represented as

$$\pi^* = \operatorname*{argmax}_{\pi} \mathop{\mathrm{E}}_{a_t \sim \pi} \left[ \sum_{t=0}^{\infty} \gamma^t \Big( r(s_t, a_t, s_{t+1}) + \alpha H(\pi(\cdot|s_t)) \Big) \right], \tag{3}$$

where $\pi$ represents the policy network, $\gamma$ represents the discounting factor, $H(\cdot)$ represents the entropy function, and $\alpha$ is the trade off coefficient.

To improve the utilization of data, the SAC maintains an experience replay buffer $\mathcal{D}$ to store historical transitions (i.e., $(s, a, r, s')$) so that the minibatch can be sampled from the buffer during training. The SAC incorporates an Actor–Critic architecture, in which the Q-network and the policy network can be updated by temporal difference and policy gradient, respectively. To tackle the problem of overestimation brought by bootstrapping, the SAC adopts the technique of target network. For faster and more stable training, the SAC exploits two Q-networks and chooses the minimum Q-value. The loss function for the Q-networks is represented as

$$L(\phi_i, \mathcal{D}) = \operatorname*{E}_{(s,a,r,s')\sim\mathcal{D}} \left[ \left( Q_{\phi_i} - y(r,s') \right)^2 \right], \tag{4}$$

where $\phi$ represents the parameters of the Q-network. The target $y$ is represented as

$$y(r,s') = r + \gamma \left( \min_{j=1,2} Q_{\phi_{targ,j}}(s', \tilde{a}') - \alpha \log \pi_\theta(\tilde{a}'|s') \right), \qquad \tilde{a}' \sim \pi_\theta(\cdot|s'), \tag{5}$$

where $\theta$ represents the parameters of the policy network, and $\tilde{a}'$ represents the next action sampled by updated policy. The loss function of the policy network is represented as

$$L(\theta, \mathcal{D}) = \operatorname*{E}_{\substack{s\sim\mathcal{D} \\ \epsilon\sim\mathcal{N}}} \left[ \alpha \log \pi_\theta(\tilde{a}_\theta(s,\epsilon)|s) - \min_{j=1,2} Q_j(s, \tilde{a}_\theta(s,\epsilon)) \right], \tag{6}$$

where $\epsilon$ represents a random number sampled from the standard normal distribution. In addition, the target networks of SAC conduct a soft update, i.e., update slowly toward the main networks in each step rather than updating periodically.

### 3.2.2. Feature Interaction-Based Policy Network and Q-Network

Feature interactions, especially high-order feature interactions, have been proven effective and efficient in modeling tabular data by many studies. In this work, we introduce a feature interaction module to extract expressive vectors from tabular data. Among dozens of feature interaction models that have emerged in recent years, GAIN is chosen by our work due to its effectiveness in learning high-order interactions and computational efficiency.

The structure of a GAIN is shown in Figure 2. GAIN takes raw features of samples as input and outputs a low-dimensional representation vector. GAIN is composed of two main modules: a cross-module and a deep module. The cross-module consists of multiple interaction units, each of which maintains a gate for every feature, and each gate only has two statuses, closed or open. Whether a feature can participate in an interaction is determined by the status of its corresponding gate. Each unit represents a feature interaction, and the interaction order is the number of gates that are open. The statuses of the gates are randomly initialized and dynamically adjusted. Through training, the statuses are gradually stabilizing, and meaningful feature interactions will be preserved. The deep module is implemented with an MLP. The outputs of the two modules concatenate to form the output of the GAIN. In our proposed model, GAIN is used as a middleware learning useful high-order feature interactions. To improve the training efficiency and reduce the parameters, the GAIN is shared by both the policy network and the Q-network.

In addition to learning higher-order feature interactions, the GAIN can also be seen as a transformer that projects the original features into a low-dimensional representation space. The benefits are twofold. On the one hand, the dimensionality reduction can avoid the curse of dimensionality. On the other hand, low-dimensional vectors facilitate the consequent distance calculation of states. In this work, the low-dimensional representation

of the state $s$ is called the abstract state, and it is denoted as $\hat{s}$. For the sake of brevity, we will not distinguish the two terms of *state* and *abstract state* in this paper unless necessary.

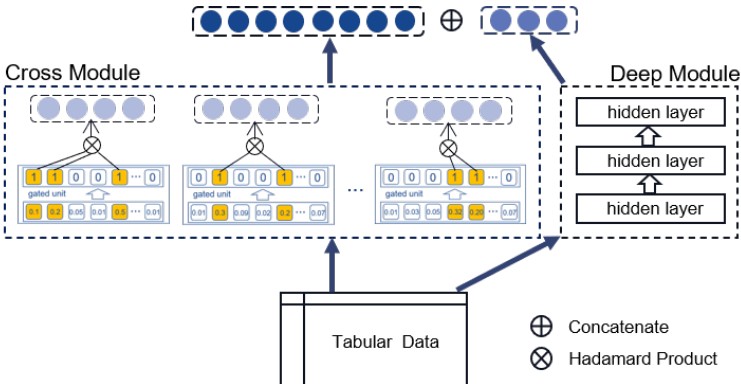

**Figure 2.** Structure of GAIN. The cross-module is composed of multiple interaction units, each of which can learn a high-order feature interaction using Hadamard product. The output of all units is concatenated with the output of the deep module to obtain the final output of the GAIN.

### 3.3. Knight Sampling Strategy

With respect to the distribution of samples, AD models typically make the following assumptions. Normal samples are large in number, and most of them are densely distributed, while anomalous samples are relatively scarce and are distributed far away from normal ones.

During the interaction between the environment and the agent, states need to be continuously sampled from the environment. The selection of a state is determined by the sampling strategy, which may affect the training efficiency. A suitable strategy could significantly improve the convergence speed of the model, while an inappropriate strategy would extend the training time or make the model converge to a local optimum. The random strategy adopted by [44] uniformly samples states from the entire state space, in which the probability of a class being selected is proportional to the proportion of states of that class in all states. Considering the severe class imbalance in the state space, normal states have much higher probabilities of being selected, and it will take more steps for the agent to encounter all anomalous states, which might result in much longer training times. To tackle the inefficient sampling of anomalous states, [45] proposes an anomaly-biased strategy, where states that are more likely to be anomalous have higher priorities to be selected. However, repeated exploitation of anomalous states would lead to overfitting of the model. Meanwhile, insufficient exploration of normal samples would also limit the model's performance.

Therefore, a novel sampling strategy needs to be devised so that all regions of the state space can be explored fully and evenly. Inspired by the Knight's Tour problem [46], we propose a novel sampling strategy. A knight-like sampler is devised that can leap to a distant state. Leaping not only make the sampler visit every part of the state space but also help the sampler escape from a cluster. An intuitive idea is to choose the farthest state from the *k*-nearest neighbors. However, considering the high computational consumption of the KNN algorithm, we take a subset instead. First, a subset $D' \subset D$ is randomly sampled; Secondly, a Euclidean distance is calculated between $s_t$ and each state in $D'$; Thirdly, the state that is farthest from $s_t$ is selected to be $s_{t+1}$. Selecting the furthest state from a random subset ensures that the entire state space can be explored and prevents the sampler from being trapped in a certain region. We name the proposed strategy as Knight Sampling Strategy.

### 3.4. Combined Reward Function

To make a trade-off between exploration and exploitation, we design a combined reward, $r_t = r_t^e + \lambda r_t^i$, where $r_t^e$ and $r_t^i$ represent extrinsic and intrinsic reward, respectively.

The former encourages the agent to exploit known information, while the latter stimulates the agent to explore novelty in the environment. $\lambda$ is the trade-off coefficient.

### 3.4.1. Extrinsic Reward Function

Extrinsic reward is an immediate reward given by the simulation environment according to the state-action pair. Since $s_t$ may come from either $D^a$ or $D^u$, the extrinsic reward function should be designed separately according to the different sources of $s_t$. If $s_t$ comes from $D^a$, a large $a_t$ should be given a positive reward. In contrast, a small $a_t$ should be given a negative reward as a penalty. Hence, $r_t^e$ should be proportional to $a_t$ (the anomaly probability) when $s_t$ is sampled from $D^u$. If $s_t$ comes from $D^u$, $r_t^e$ is inversely proportional to $a_t$. Considering the possible anomalies in $D^u$, to prevent the agent from easily identifying a sample from $D^u$ as normal, we should give a minor reward if $a_t \to 0$. Similarly, if $a_t \to 1$, a minor penalty is more feasible since a large penalty would discourage the agent from detecting hidden anomalies. A coefficient, $\tau$, is utilized to scale the reward, and the value of $\tau$ is usually specified as the ratio of the sizes of $D_a$ and $D_u$. To be compatible with the continuous action space, a continuous extrinsic reward function is required, which is defined below.

$$r_t^e(s_t, y_t, a_t) = \begin{cases} 2a_t - 1 & \text{if } y_t = 1, \\ -\tau \cdot (2a_t - 1) & \text{if } y_t = 0 \end{cases} \tag{7}$$

### 3.4.2. Intrinsic Reward Function

Inspired by human experiences in playing games that the highest score can only be obtained if the environment is fully explored, we design a curiosity-driven intrinsic reward function to encourage the agent to explore states with high novelty. Intuitively, the novelty of a state will decrease if it is sampled several times. In addition, the same happens with repeated sampling of nearby states. Therefore, the intrinsic reward of a state-action pair is inversely proportional to the visits to the region where the state is located. We exploit a Gaussian kernel function to approximately calculate the number of visits, which is represented as

$$\kappa(x, y) = exp-\frac{\|x - y\|_2}{2\sigma^2}, \tag{8}$$

where $\| \cdot \|$ returns a Euclidean distance, and $\sigma$ is a hyper-parameter, which is discussed in Section 5.6. If two states are close to each other, $\kappa$ tends to be 1. In contrast, if two states are far away from each other, $\kappa$ tends to be 0.

We define an episodic memory $M = [\hat{s}_1, \hat{s}_2, \cdots, \hat{s}_{t-1}]$ to store the states before time step $t$. Since the kernel function can convert the distance of two states between 0 and 1, we calculate $\kappa$ of $\hat{s}_t$ and each state in $M$ and take the reduction sum as the approximate counts. The intrinsic reward function is represented as:

$$r_t^i = \frac{1}{\sqrt{\sum_{i=1}^{n-1} \kappa(\hat{s}_t, \hat{s}_i) + 1}}. \tag{9}$$

## 4. Model Analysis

### 4.1. Analysis of the Agent

Our proposed model exploits a DRL framework to solve the TAD problem. We make some adaptations to the vanilla SAC by replacing the MLP in the Actor and the Critic with a GAIN that can generate more expressive representations. The output of the Actor can be used to express the uncertainty of the agent's judgment, i.e., $a \to 0$ or $a \to 1$ indicates that the agent is quite confident in its judgment, while $a \to 0.5$ indicates the opposite. In the early stages of training, the model has not learned enough information to make the best decision. Hence the uncertainty should be preserved to avoid premature convergence. The entropy regularization term aims to enhance the exploration of actions, and a coefficient $\alpha$ is used to make a trade-off between expected rewards and entropy. As training continues,

the uncertainty will decrease. To accelerate the convergence of the model, the value of $\alpha$ gradually reduces.

### 4.2. Explanation of the Knight Sampling Strategy

Because there is no environment that can automatically generate a new state according to the agent's action, we create a simulation environment that returns a state and a reward to the agent at each time step. There exist two basic facts about the anomaly detection task. First, the number of normal states in the training set far exceeds the number of anomalous ones. Secondly, normal states are relatively densely distributed, while anomalous states are sparsely distributed and far from normal ones. Based on the facts above, we propose a knight sampling strategy to sample evenly from both categories of states. The knight sampling strategy, which can be understood as dividing the state space into a chessboard with a grid, will alleviate the problem of severe category imbalance. The distribution of 100 states is shown in Figure 3; normal and anomalous states are indicated by a blue dot and red cross, respectively. The ratio of anomalous states to normal ones is 1/9. After dividing the state space with a grid, the ratio of anomalous squares to the normal ones increases to 9/17. Guided by this strategy, the sampler leaps around the data space like a knight, and each move returns a state from the current square.

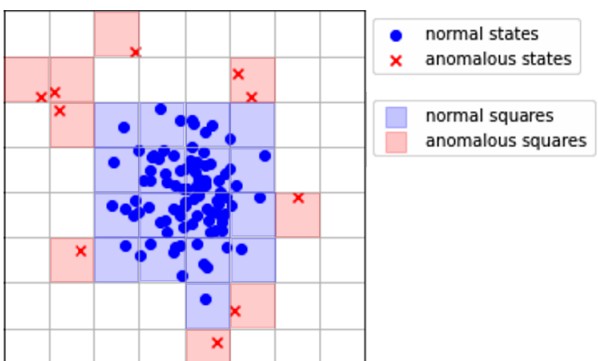

**Figure 3.** Explanation of Knight Sampling Strategy. Our proposed strategy gives the sampler the leaping capability of a knight and divides the entire data space into grids. The leap capability allows the sampler to fully and uniformly sample from the data space by traversing all the squares.

### 4.3. Combined Reward

In addition, we deliberately design a combined reward function to make a trade-off between exploitation and exploration of the data space. The extrinsic reward function leverages the continuous reward function to return a reward for each action to settle the sparsity of the rewards. The continuous reward function can be regarded as a generalization of the discrete reward function over the continuous action space. From the perspective of Information Theory, the intrinsic reward aims to encourage the agent to explore the states that are expected to bring high information gains. Although all states contain information, the information gain will decrease when a state is sampled repeatedly. The Gaussian kernel function, which is exploited to approximately count the times of sampling, returns the similarity of the current state and the previous ones. Due to the nature of the exponential function, a few states that are quite similar to the current state will be counted. To cope with this problem, we can adjust the hyper-parameter $\sigma$ to control the counting radius. The similarities between all points and the center in a square are calculated using Equation (8), and the results are shown in Figure 4. The four subgraphs represent the influence of $\sigma$ on the distribution of similarities.

We can tell from Figure 4 that if $\sigma$ is assigned a small value, the similarity decreases sharply with the increase in distance from the center. Hence most neighbors of the center are omitted as they share a similarity close to 0. Therefore, only a few historical states that are very close to the center will be counted. On the contrary, if $\sigma$ is assigned a

larger value, the similarity decreases slowly. Hence more states make contributions to Equation (9), which indicates that more historical states will be counted. Consequently, the hyper-parameter $\sigma$ can be approximated as the counting radius.

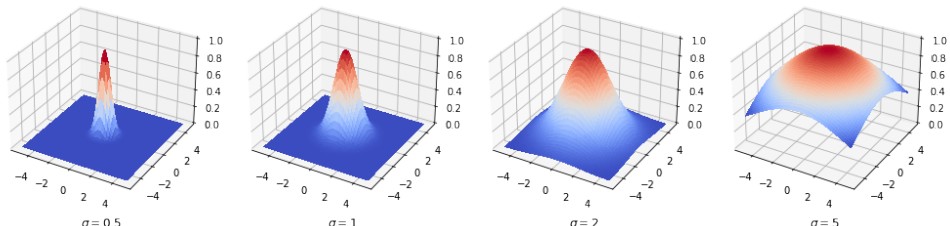

**Figure 4.** Effect of $\sigma$ on similarity. The four subplots represent the Gaussian kernel function surfaces of the surrounding points and the center point for different values of the $\sigma$. The closer the color is to red, the closer the similarity is to 1. The closer the color is to blue, the closer the similarity is to 0. As the $\sigma$ increases, the steep surface gradually becomes flatter, and the red area gradually increases, which means that the number of points similar to the center gradually increases.

## 5. Experiments

### 5.1. Datasets

To verify the validity of our proposed model, three datasets from different application scenarios are selected for our experiments. NSL-KDD is a dataset from the domain of web security, and each sample in it represents a network traffic record that consists of 40 features. The value of *normal* in feature *attack_type* indicates benign connections, while other values indicate malicious ones. Credit card is a dataset from the domain of finance, which contains credit card transactions conducted by cardholders in Europe over two days in September 2013. Each sample in the dataset represents a transaction record, which consists of 30 features. The value of 1 in *Class* indicates an anomalous transaction, while the value of 0 indicates the opposite. Census is a dataset from the domain of sociology, which contains weighted data extracted from the 1994 and 1995 current population surveys conducted by the U.S. Census Bureau. The dataset contains 40 features, including demographic and employment information, in which the record with *income* "50,000+." are regarded as anomalies. The details of the three datasets are listed in Table 1.

**Table 1.** Datasets Details. "num." and "cat." denote numerical features and categorical features, respectively.

| Dataset | # Samples | # Features | % Anomaly |
|---------|-----------|------------|-----------|
| NLS-KDD | 148,517 | 7 cat. + 33 num. | 48.12 |
| Credit Card | 284,807 | 30 num. | 0.17 |
| Census | 299,285 | 33 cat. + 7 num. | 6.20 |

The three datasets are very representative for their different composition of features, i.e., NSL-KDD contains 7 categorical features and 33 numerical features, census contains 33 categorical features and 7 numerical features, and credit card only contains numerical features. Recall the assumptions in Section 3.1 that a dataset for AD task consists of a small labeled subset of anomalies and a large-scale unlabeled subset. To make the dataset of NSL-KDD fit our assumption, a downsampling technique is employed to reduce the anomaly ratio. NSL-KDD contains four types of anomalies, each with a large variation in sample size, i.e., *dos*: 45927, *r2l*: 11656, *probe*: 995, *u2r*: 52. Considering the severe intra-imbalance of anomalies, we keep all samples from *probe* and *u2r*, and retain only 11–12% samples from *dos* and *r2l*. The downsampling operation reduces the anomaly ratio from 48.12% to 10.06%.

We select six competing methods to perform a performance comparison with our proposed model. In their official implementations, raw features are not preprocessed in

the same manner, e.g., iForest directly removes all categorical features, CatBoost converts categorical features to continuous values, Deep SAD transforms all features to a low-dimensional vector, FT-Transformer transforms all features to embedding vectors with the same length, etc. To reduce the impact of different preprocessing methods on performance, we use the same data preprocessing for all models. First, numerical features are discretized into categorical features; Secondly, all categorical features are embedded into a vector with the same length.

### 5.2. Competing Methods

Dozens of methods have been proposed to tackle the TAD problems in recent years. According to the difference in the leveraging of supervisory information, those methods can be divided into four categories, such as unsupervised learning methods (UN), supervised learning methods (SU), semi-supervised learning methods (SS) and reinforcement learning methods. In this work, two state-of-the-art methods from each category are selected as the baselines.

- **iForest** [12]. iForest (UN) determines a sample's anomaly degree based on the distribution of each feature value in its feature field. It assumes that the feature values of anomalies are sparsely distributed and, therefore, can be easily distinguished from that of normal samples. A split tree is built for each feature field, and the depth of the feature value represents the anomaly score of a single feature. By combining the anomaly scores of multiple features, the anomaly score of a sample can be obtained.
- **CBLOF** [47]. CBLOF (UN) is a cluster-based anomaly detection method that assumes that anomalies count for a small proportion of the total size and that the samples far away from the large cluster can be considered anomalies. First, a clustering method (e.g., K-Means clustering) is used to cluster samples; Secondly, large clusters are distinguished from small ones; Lastly, distances between samples and large clusters are calculated as anomaly scores.
- **CatBoost** [48]. CatBoost (SU) uses gradient boosting on decision trees. It is an ensemble algorithm that creates a strong learner from an ensemble of multiple weak learners. As its name suggests, CatBoost is capable of handling categorical data. In addition, it solves the problems of gradient bias and prediction shift.
- **FT-Transformer** [24]. FT-Transformer (SU) is a deep model that adapts transformer architecture to tabular data. All features are first transformed to an embedding vector and then passed to a stack of transformer layers to get the prediction.
- **Deep SAD** [18]. Deep SAD (SS) tries to learn a neural network that maps samples to a low-dimensional space in which normal samples cluster in a compact hypersphere while anomalous ones are located outside the hypersphere. A few labeled anomalies can be used in Deep SAD to improve the model's predictive accuracy.
- **DevNet** [20]. DevNet (SS) is an end-to-end semi-supervised method that leverages a limited number of labeled anomalies as prior knowledge to predict anomaly scores. It is based on the assumption that there exists significant statistical deviation between normal samples and anomalous ones.

### 5.3. Evaluation Metrics

Two popular and complementary metrics, AUC-ROC (Area Under Receiver Operating Characteristic Curve) and AUC-PR (Area Under Precision-Recall Curve), are chosen as the evaluation metrics in this work. AUC-ROC, which summarizes the ROC curve of true positives against false positives, is used to evaluate the classification performance of a model. However, AUC-ROC cannot truly reflect the classification performance when the classes of samples are severely imbalanced. Whereas AUC-PR, which summarizes the ROC curve of precision against the recall and focuses on the performance of anomaly class, is more suitable for this work. A larger AUC-ROC or AUC-PR value reflects better performance.

*5.4. Performance Comparison with Competing Models*

To verify the effectiveness of our proposed model, comparative experiments are conducted on the three datasets, and the results are presented in Table 2. Our proposed FIR-TAD performs consistently better than its competitors no matter in datasets with more categorical features or in the dataset with more numerical features. In addition, several observations can be obtained from the results. First, unsupervised methods cannot outperform supervised or semi-supervised methods. On the one hand, the absence of labels reduces the information available. On the other hand, unsupervised methods are usually based on the assumption that anomalous samples are significantly different from normal ones in distribution, which may not provide sufficient accuracy. Secondly, semi-supervised methods perform slightly better than supervised methods due to the exploitation of labeled anomalies. Thirdly, the DRL methods show strong competitiveness compared with other deep methods, especially FIR-TAD achieves substantially better performance. We attribute this to the introduction of feature interactions and the exploration ability of SAC.

**Table 2.** Performance comparison.

| Model | NSL-KDD | | Credit Card | | Census | |
|---|---|---|---|---|---|---|
| | AUC_ROC | AUC_PR | AUC_ROC | AUC_PR | AUC_ROC | AUC_PR |
| iForest | 0.8359 | 0.8483 | 0.9469 | 0.1412 | 0.5956 | 0.0781 |
| CBLOF | 0.8213 | 0.5307 | 0.8772 | 0.2451 | 0.5938 | 0.0742 |
| CatBoost | 0.9208 | 0.9012 | 0.8446 | 0.4867 | 0.8815 | 0.3510 |
| FT-Transformer | 0.9278 | 0.8530 | 0.8275 | 0.4499 | 0.8376 | 0.2335 |
| DeepSAD | 0.9391 | 0.9037 | 0.8902 | 0.2577 | 0.7232 | 0.1855 |
| DevNet | 0.9410 | 0.9278 | 0.9520 | 0.5109 | 0.8354 | 0.3211 |
| FIRTAD(ours) | 0.9457 | 0.9362 | 0.9583 | 0.5870 | 0.8952 | 0.3670 |

*5.5. Test on Robustness*

Robustness with regard to anomaly ratio. To study the robustness of all models on datasets containing different numbers of anomalies, we create five datasets with different proportions (10%, 5%, 1%, 0.5%, 0.1%) of anomalies based on the NSL-KDD. Comparative experiments are performed on the five datasets, and the results are shown in Figure 5a. We choose AUC-PR as the only evaluation metric in the following because the results of AUC-ROC are often over-optimistic in the case of imbalanced classes. As shown in Figure 5a, the performance of all models degrades with the decrease in the proportion of anomalies, and the unsupervised methods are less affected as they distinguish anomalies based on differences in the distribution of normal samples and are not sensitive to the number of anomalous samples. With the reduction in supervised information, the supervised methods experience significant performance degradation due to their reliance on supervisory information. Although the semi-supervised methods are less affected by the decrease in the ratio of anomalies, we note that the performance of the semi-supervised methods is significantly weaker than that of the unsupervised method when the ratio drops below 0.5%. In contrast, our proposed method shows better robustness in all cases.

Robustness with regard to label contamination. To test the robustness of all models under label-contaminated conditions, which is very common in real-world scenarios, we define five datasets by sampling incremental numbers of labeled anomalies, removing their labels and blending them into the unlabeled samples. The contamination rates of the five datasets are 0%, 2%, 5%, 10% and 20%, respectively. All models are evaluated on these five datasets, and the experimental results in Figure 5b show that the increase in label contamination rate has almost no impact on the unsupervised method. The supervised methods suffer the most due to the decrease in data quality. The semi-supervised methods consistently perform better than the unsupervised methods, proving their good robustness to label contamination. Our proposed model exhibits the best robustness, and we attribute

it to the design of the extrinsic reward function, which encourages the exploration of anomalies in normal samples.

Robustness with regard to unknown anomalies. For some online anomaly detection systems, the training and test sets usually have inconsistent sample distributions, i.e., the test set may contain anomalies that never appear in the training set. To compare the robustness of different methods in the face of unknown anomalies, we define a dataset in which the training and test set contains only one type of anomaly and four additional test sets by adding new anomaly types to the test set in turn. We evaluate the trained model on the five test sets and present the results in Figure 5c. The performance of all models decreases with the increase in anomaly types, with the unsupervised methods being the least affected and the supervised methods being the most affected. The semi-supervised methods still exhibit good robustness. Our proposed model performs significantly better than its competitors. The introduction of feature interaction allows our model to learn useful discriminative information, and the exploitation of the DRL algorithm gives our model the ability to explore the unknown. These two factors give our model very good robustness to unknown anomalies.

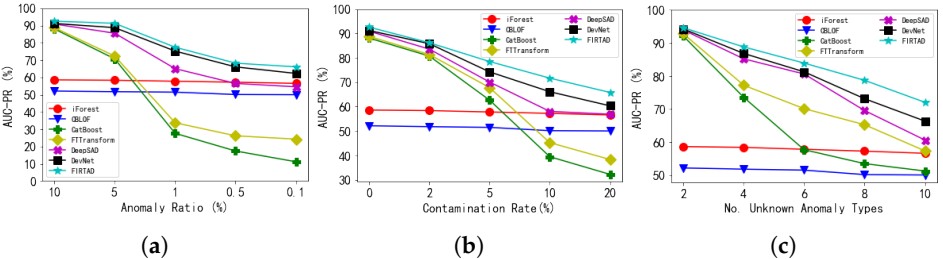

**(a)**          **(b)**          **(c)**

**Figure 5.** Test on Robustness. (**a**) Robustness with regard to anomaly ratio; (**b**) Robustness with regard to label contamination; (**c**) Robustness with regard to unknown anomalies. We find that although the unsupervised models exhibit the best robustness, their performance is limited. The supervised models suffer most from the decrease in supervised information and label quality; they perform even worse than unsupervised models. The semi-supervised models show good robustness against the anomaly ratio. However, the performance degrades significantly with respect to label contamination and unknown anomalies. Our proposed FIRTAD shows better robustness and consistently outperforms the other models.

### 5.6. Impacts of Hyperparameters

In this section, we mainly focus on the impacts of the hyperparameters using the NSL-KDD dataset.

- Impacts of representation dimensionality of each feature ($d$). In a deep model, the representation vector with a longer length carries more information; hence, it is common to improve the expressiveness of a model by increasing the representation dimensionality of the input. Nevertheless, it is a double-edged sword, as an increase in the dimension of the representation vector would lead to an increase in memory consumption and a decrease in model efficiency. We choose $d = 4, 8, 16, 32$ and plot the AUC-PR in Figure 6a. Apparently, the performance improves as the length of the vector increases. Concretely, when $d$ changes from 8 to 16, the AUC-PR improves from 0.9175 to 0.9437, with an improvement of 2.86%. However, as the length of the vector continues to increase, the improvement in model performance becomes very limited. The AUC-PR improves from 0.9437 to 0.9439 when $d$ increases from 16 to 32. Considering the consequent doubling of memory consumption and computation time, $d = 16$ seems to be a more reasonable option, which creates a balance between effectiveness and efficiency.

- Impacts of coefficient of intrinsic reward ($\lambda$). As discussed in Section 3.4, $\lambda$ is leveraged to provide a tradeoff between exploitation and exploration, which represents the

weight factor corresponding to the intrinsic reward. We choose $\lambda$ from $0, 0.3, 0.5, 0.8, 1$, in which different values correspond to different extensions of exploration, e.g., $\lambda = 0$ represents a deprecation of exploration, while $\lambda = 1$ represents the opposite. The experimental results are plotted in Figure 6b. We can tell from the results that the introduction of intrinsic rewards does improve the model's performance, while a high weight may cause the model to converge prematurely to suboptimal solutions. As shown in Figure 6b, $\lambda = 0.5$ is the most suitable option.

- Impacts of counting radius ($\sigma$). To encourage the agent to explore unknown regions in the environment, we designed the intrinsic reward to score the novelty of a region. The number of samplings from a region is approximately counted by a Gaussian kernel function, in which $\sigma$ can be regarded as the counting radius. For a specific region that has been visited a certain number of times, increasing the radius will lead to an increase in novelty and vice versa. We choose the value of $\sigma$ from $0.1, 0.5, 1, 2, 5$ and plot the results in Figure 6c. Intrinsically, a small radius would result in large intrinsic rewards for new samples, even if similar ones have been sampled many times, which might hinder the agent from exploring unknown regions. Moreover, a large radius would result in small intrinsic rewards for samples from sparse regions, which might result in insufficient exploration of known regions. Both of the above conditions would reduce convergence speed or make the model converge to a suboptimum, and this intuition is verified by the experimental results. As shown in Figure 6c, $\sigma = 1$ is the most reasonable option.

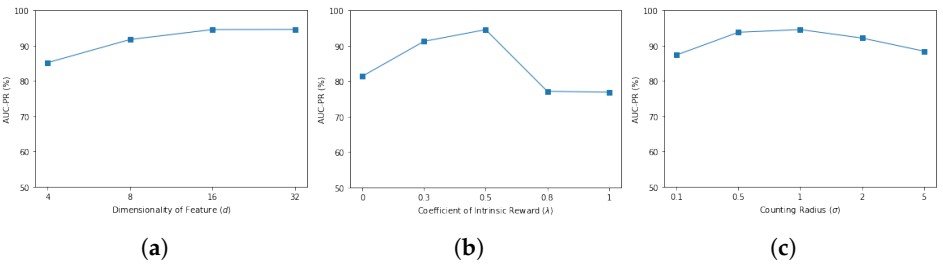

|  (a)  |  (b)  |  (c)  |

**Figure 6.** Impacts of Hyperparameters. (**a**) The contribution of representation dimensionality to model performance improvement decreases with the increase in dimension; (**b**) The introduction of intrinsic reward improves the model's performance, but a high weight would be counterproductive; (**c**) A small counting radius will increase the intrinsic reward and thus lower the agent's desire to explore the state space, while a large counting radius will render a small intrinsic reward and hence lead to insufficient exploration of states.

*5.7. Ablation Study*

To investigate the effects of different components in our proposed model, we propose several variants based on FIRTAD and conduct ablation experiments on all three datasets.

1. MLP-SAC. MLP-SAC replaces the feature interaction module (GAIN) with an MLP. To reduce the number of parameters and accelerate model convergence, the MLP is shared by the Actor and Critic.
2. FIR-DDPG. FIR-DDPG replaces the SAC with the Deep Deterministic Policy Gradient (DDPG), which is an Actor–Critic, model-free algorithm based on the deterministic policy gradient that can operate over continuous action spaces.
3. FIR-AB. FIR-AB replaces the knight sampling strategy with the anomaly-biased strategy used in [45].

Table 3 shows the results of our original model, FIRTAD, and its variants. In the following, we analyze the effects of components in our model. First, compared to the original model, MLP-SAC shows a significant drop in performance on NSL-KDD and census. This implies that the feature interaction module indeed enhances our model's ability for anomaly detection. Further, we note that MLP-SAC achieves comparable results

to the original model on credit card, suggesting that GAIN has no advantage in dealing with data consisting entirely of numerical features. Secondly, although both SAC and DDPG are off-policy algorithms for continuous action space, SAC has a stronger exploratory capability than DDPG due to its exploitation of stochastic policy. The comparison result between FIR-DDPG and FIRTAD validates this view and prove the effectiveness of the SAC module. Thirdly, the performance gaps between FIR-AB and FIRTAD in the three datasets indicate that the knight sampling strategy may outperform the anomaly-biased sampling strategy.

**Table 3.** AUC-PR Performance of FIRTAD and its Three Ablated Variants.

| Datasets | FIRTAD | MLP-SAC | FIR-DDPG | FIR-AB |
|---|---|---|---|---|
| NLS-KDD | 0.9362 | 0.6874 | 0.9012 | 0.8829 |
| Credit Card | 0.5870 | 0.5798 | 0.5022 | 0.3681 |
| Census | 0.3670 | 0.1453 | 0.3124 | 0.2445 |

## 6. Conclusions

In this paper, we propose a novel anomaly detection method for tabular data called FIRTAD, which incorporates feature interaction techniques into a deep reinforcement learning framework. The innovative aspects of this article are manifested in the following dimensions: (1) It is an anomaly detection system specifically designed for tabular data; (2) It employs the SAC algorithm to generalize the discrete action space into a continuous space, thereby enhancing the model's expressive power; (3) It creates a simulation environment by devising a novel sampling strategy and a combined reward function; (4) As far as we know, it is the first effort to apply feature interaction to anomaly detection tasks. The experiments demonstrate that our proposed model not only outperforms state-of-the-art models in terms of performance but also exhibits good robustness in situations involving varying anomaly ratios, label contamination and unknown anomalies. Our model is applicable to real-world anomaly detection scenarios, particularly in domains that have accumulated some anomalies. This work serves as an attempt to apply deep reinforcement learning to anomaly detection task and may provide some inspiration to relevant researchers.

Despite the encouraging results, our proposed FIRTAD still has some limitations. First, our model relies heavily on large amounts of data for training, so its performance advantages may not be apparent when only dealing with a small dataset. Second, the anomaly detection capability of our model gradually improves during the interaction between the agent and the environment, which leads to an increased demand for computational resources and training time. Third, the experiments demonstrate that the performance advantages of our model on the balanced dataset are not significant, which limits the applicability of the model.

Future work will consider using multiprocessing for model training and the study of anomaly interpretability.

**Author Contributions:** Conceptualization, Y.L. and L.M.; methodology and implementation, Y.L. and S.Z.; writing—original draft preparation, Y.L.; writing—review and editing, L.M., M.W. and S.Z. All authors have read and agreed to the published version of the manuscript.

**Funding:** This research received no external funding.

**Institutional Review Board Statement:** Not applicable.

**Informed Consent Statement:** Not applicable.

**Data Availability Statement:** The NSL-KDD dataset supporting this study was obtained from https://www.unb.ca/cic/datasets/nsl.html (accessed on 13 February 2023). The credit card dataset was obtained from https://www.kaggle.com/mlg-ulb/creditcardfraud (accessed on 13 February 2023). The census dataset was obtained from http://archive.ics.uci.edu/ml/machine-learning-databases/census-income-mld/ (accessed on 13 February 2023).

**Acknowledgments:** The authors would like to thank the anonymous reviewers for their constructive comments and suggestions.

**Conflicts of Interest:** The authors declare no conflict of interest.

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
