# Peer review of "Feature Interaction-Based Reinforcement Learning for Tabular Anomaly Detection"

_electronics, doi:10.3390/electronics12061313_

Round 1

Reviewer 1 Report

   The manuscript by Liu et al. is devoted to development of new method of anomaly detection. This work seems to be novel and interesting; I have only one minor comment: Caption of figures should be extended because they are not understood without text. I suppose that minimal understanding of figures without text (on basis of captions only) can make this work more suitable for potential readers.

Reviewer 2 Report

The paper suggests a DRL-based anomaly detection technique particularly for tabular data. The technique formulates a simulation environment which lets all values be surveyed.

The paper is nice and has a decent contribution; however, I have several remarks:

1. In section 2.2 the authors mention that [33] suggests a parallel scheme. The parallel approach for learning was also suggested in S. T. Klein & Y. Wiseman, "Parallel Lempel Ziv Coding", Journal of Discrete Applied Mathematics, Vol. 146(2), pp. 180-191, 2005. Available online at: https://u.cs.biu.ac.il/~wisemay/dam2005.pdf and also in Nakasato, N., "Implementation of a parallel tree method on a GPU", Journal of Computational Science, Vol. 3(3), pp. 132-141, 2012. I would encourage the authors to cite these papers and explain that a parallel approach can enhance their processing times.

2. The equations in section 2.3 should be numbered.

3. The authors do not define r^i_t and do not use it in the equations, although it appears in the text.

4. In section 3, the authors write about "small subset" and "large scale subset". This is not well-defined. What are the sizes of your subsets?

5. Lambda in section 3.1 is not well defined. Is it a number between 0 to 1?

6. Equation 1 in section 3.2: many letters are not explained.

7. Equation 3 and equation 4, I did not find an explanation why the authors took the log of PI.

8. In Table 1, the authors write "To make the three datasets fit our assumptions, a downsampling technique is employed to the dataset of NSL-KDD to reduce the ratio of anomalies from 48.12% to 10.06%." This should be explained further. Did the authors have manipulated the data?

Reviewer 3 Report

The article is well written. Some suggestions are -

(i) add more related work form 2021 and 2022,

(ii) add limitation  of the work

(Iii) elaborate a bit more on the novelty and significance of the work 

Round 2

Reviewer 2 Report

The authors made a decent effort and the paper is certainly publishable so I would recommend accepting the paper.